# Religious Experience and Yoga

**Christopher Key Chapple**

Loyola Marymount University, 1 Loyola Marymount University Dr., Los Angeles, CA 90045, USA; cchapple@lmu.edu

**Abstract:** Yoga practice provides access to religious experience, which has been defined by William James as "immediate luminousness, philosophical reasonableness, and moral helpfulness." In this paper the processes of Yoga will be summarized as found in the *Bhagavad Gītā* and the *Yoga Sūtra*. This article concludes with instructions on how to perform a practice that integrates Yoga breathing and movement with reflections on the Sāṃkhya descriptions of physical and emotional realities (*tattvas* and *bhāvas*).

**Keywords:** William James, Yoga, Gāyatrī mantra, Bhagavad Gītā, Patañjali, Sāṃkhya philosophy, Gandhi, Bhakti, Vedānta, *Yoga Sūtra*

---

## 1. Yoga and the Light

As one enters a conversation about religious experience, Yoga and Hinduism, it can be helpful to begin with images of light. Vedic mantras signaling the importance of light include the Pavamāna and Gāyatrī mantras.

> *asato mā sad gamaya,*
> *tamaso mā jyotir gamaya,*
> *mṛtyor mā amṛtaṃ gamaya,*
> Lead me from chaos into order.
> Lead me from lethargy into the light.
> Lead me from death to immortality.
> —Bṛhadāraṇyaka Upaniṣad 1.3.28[1]

> *tatsaviturvareṇyaṃ*
> *bhargo devasyadhīmahi*
> *dhiyo yo naḥ prachodayāt*
> The sun has no rival.
> We reflect on the luminosity of its power.
> May it inspire our own vision.
> —Ṛg Veda 3.62.10

The term light (*jyotir*) in the Pavamāna Mantra and the invocation of the rising sun (*savitur*) in the Gāyatrī Mantra both indicate a unity between the internal process of waking up the human body and the external, macrocosmic event of sunrise. Both these mantras are frequently chanted in the world of modern Yoga and lend a sense of antiquity and continuity to the contemporary practice of what Elizabeth deMichelis has called "Modern Postural Yoga."[2] The implication of both is clear: a light can be kindled within through processes of Yoga and meditation. This inner light mirrors and

---

[1] All translations are by the author.
[2] (De Michelis 2004).

connects with the power of the rising sun. To become an adept at Yoga entails moving towards light and away from darkness, to arrive at a place of spiritual enlightenment.

Two other light-referent terms central to Yoga are *sattva* and *samādhi*. The former indicates the lightest state of being that comes closest to replicating the luminosity of witness consciousness, the seer (*puruṣa*, *draṣṭṛ*, *Yoga Sūtra* II:41, III:35, 49, 55); while the latter indicates full emplacement within the state of being completely absorbed, ego-free, and free (*YS* I:20, 46, 51; II:2, 29, 45; III:3, 11, 37; IV:1, 29). Both terms indicate a state of being filled with light and lightness, no longer weighed down by the effects of past karma. Such a person is free from regrets about the past as well as content in terms of what might happen in the future.

In Hindu Yoga traditions, this experience of lightening becomes externalized and internalized, observed and witnessed as well as felt in the realm of affect. Aesthetic moments can stun a person into a silent state, a direct connection with beauty and awe. Two practices enhance the possibility of this experience: seeing images of the divine in a statue or in a living exemplar (*darśana*) and the performance of rituals that create a mood of reverence. External rituals can be elaborate Vedic sacrifices (*yajña*), simple home devotionals (*pūjā*), the veneration of a teacher (*guru-śraddhā*), or pilgrimage to a temple (*mandir*) or some other holy place (*tīrtha*).[3] Internal rituals that kindle the inner light (*jyotir*) include the practice of various forms of Yoga, including reflective attempts at self-improvement, bodily movement to generate heat (*tapas*) that purifies the body (*śuddhi-śarīra*), breath control, developing a sense of inwardness leading to concentration and meditation, culminating with the still of the minding into a state of absorption. Quite often this practice will be coupled with the more external devotions mentioned above, and the recitation of mantra and singing.

## 2. Yoga and Religious Experience: James and Gītā

Juxtaposing the words Yoga and religious experience, one automatically goes to William James and his book *Varieties of Religious Experience* (1902). This seminal work in many ways places Yoga at the nexus of conceptualizing a religious experience, in terms of both process and actualization. James posits three criteria for assessing genuine religious experience, which he places in italics. They include: "*immediate luminousness … philosophical reasonableness*, and *moral helpfulness*"[4]. This article will explore what can be expected and achieved within these three categories through Yoga.

What is Yoga? Patañjali, in the early centuries of the common era, defined Yoga as *citta-vṛtti-nirodhaḥ*, the restraint of mental fluctuations (YS I:2). Gurāṇi Añjali (1935–2001), founder of Yoga Anand Ashram, proclaimed that "Yoga is a point in time where a sacred secret occurs. And the individual is filled with an ecstasy that stops all language."[5] This latter definition somewhat resembles William James's definition of Yoga as "training in mystical insight that has been known from time immemorial."[6] James, in his description of Yoga, quotes Swami Vivekananda's *Raja Yoga*: "There is no feeling of I, and yet the mind works, desireless, free from restlessness, objectless, bodiless. Then the Truth shines in its full effulgence."[7] From darkness, one has turned towards light.

Perhaps one of the best places to assess Yoga in terms of James's three criteria of *immediate luminousness… philosophical reasonableness*, and *moral helpfulness* would be the four forms of Yoga articulated in the Bhagavad Gītā: discernment or Jñāna Yoga, action or Karma Yoga, devotion or Bhakti Yoga, and meditation or Raja Yoga. It is only after great struggle that Arjuna, the protagonist of the Gītā, comes into a state of *luminousness*, albeit fleeting, following instruction in meditation and devotion. From the start of the text, Arjuna's preceptor Krishna brings him to a place of *philosophical reasonableness* by instructing him in the physical and metaphysical teachings of Sāṃkhya and Vedānta philosophies, through Jñāna or discernment Yoga. In various ways, Krishna instructs Arjuna on the complexities of *moral helpfulness*, specifying that Karma Yoga, with its sense of aplomb, will see one

---

[3]  See the works of C. J. Fuller, Axel Michaels, Diana Eck, Constantina Rhodes, James J. Preston and others for full descriptions of these practices.

[4]  (James [1902] 1961).

[5]  (Añjali forthcoming).

[6]  James, op. cit. p. 314.

[7]  Vivekananda as quoted in James, p. 325.

through even the most difficult of tasks, and that it is possible to hold to one's dignity whether faced with humiliation or glory.

The Yoga of the Bhagavad Gītā begins with a crisis. Arjuna, faced with the prospect of slaying family members and teachers on the Kurukshetra battlefield, falls into a state of paralysis. On their shared chariot, his cousin, Avatāra Krishna, instructs Arjuna about the ways of discernment (*jñāna*) and steady wisdom (*sthita prajñā*).

Gandhi discovered the *Bhagavad Gītā* while in England. For him it became the touchstone to states of Yoga. Through its narrative he discovered a way to think more expansively about his own story. He sought solace particularly in the last eighteen verses of the second chapter (54–72), finding inspiration in their message: be the best person you can possibly be, at all times and in all circumstances. To understand Yoga as a tool of reasonableness and moral helpfulness, this section of the text will be considered, as well as other passages that similarly describe that exemplary person who is able to maintain dignity and calm in the midst of chaos and difficulty.

Arjuna asks Krishna:

> How can the person of steady wisdom be described,
> that one accomplished in deep meditation?
> How does the person of steady vision speak?
> How does such a one sit and even move?
> The Blessed One responds:
> When a person leaves behind all desires
> that arise in the mind, Arjuna,
> and is contented in the Self with the Self,
> that one is said to be steady in wisdom.
> The person who is not agitated by suffering (*duḥkha*),
> whose yearnings for pleasures have evaporated,
> whose passions, fear, and anger have evaporated,
> that sage, it is said, has become steady in vision.
> One whose passions have been quelled on all sides
> whether encountering anything, whether pleasant or unpleasant,
> who neither rejoices or recoils,
> such a person is established in wisdom.
> And when this person can draw away from the objects of sense
> by recognizing the senses themselves
> like a tortoise who draws in all five of its limbs,
> such a person is established in wisdom.

Krishna explains how restraint of the senses allows stability, and then describes how attachment and the blind pursuit of desires can lead to one's downfall:

> Fixation on objects
> generates attachment.
> Attachment generates desire.
> Desire generates anger.
> Anger generates delusion.
> From delusion, mindfulness wanders.
> From wandering mindfulness arises the loss of one's intelligence.
> From the loss of intelligence, one perishes.
> By giving up desire and hatred
> even in the midst of the sense objects
> through the control of the self by oneself,
> a person attains peace. (Translation of BG II:54–72 by the author.)

Krishna tells Arjuna that this peace equips a person with the discipline needed to practice meditation and that "Without meditation there can be no tranquility. Without tranquility, how can

there be happiness?" An adept with the stabilized mind, grounded in peace, and skilled in meditation is described as one "free from possessiveness, free from ego". These qualities encapsulate the best of what is possible through Yoga. This section of the *Gītā* provides a definition of Yoga in accord with the Jamesian principles of reason and helpfulness. Krishna urges Arjuna to cultivate a way of being in oneself and in the world that does not fall prey to distraction, desires, and selfishness. Holding steady, one is able to cleave to what is most central and dispel all forms of delusion.

This poem-within-a-poem can be parsed into four basic messages, starting with the initial volley of Arjuna's question. Arjuna has been utterly paralyzed by his situation. He feels miserable, defeated, confused, and impotent. His world has been so radically shaken by treachery committed by his own cousin-brothers that he cannot move forward. The first message lies in the opening question: we must look for a way of being in the world that will provide peace and tranquility.

The second message of this *Gītā* portion asks for a reconsideration of the fixity of the external world. The external world "arrives" because we say it is so, because of agreed-upon conventions about right and wrong, tasty and disgusting, worthy and unworthy. Krishna provides a measured critique and analysis of this habitual way of engaging with the world. He calls into question the relationship between the senses and the objects of the senses. Krishna urges one to "dial it back", to recognize that a sense object does not exist before the sensory organ (*indriya*) "lands" upon it, seizes it, and makes it real. Careful direction of the senses can help shape one's emotional relationship with the world. By learning to step back into a place of consideration before, in Nietzsche's words "going under," in this case under the thrall of the senses, one can gain a measure of mastery that ultimately leads to self-understanding and self-control. Releasing the grip of what one wishes to be, one can face reality and respond accordingly.

Third, Krishna articulates a cascade of unfortunate consequences that can result if one does not gain self-control. Attachment leads to desire. Thwarted desire leads to anger. Anger confuses the mind. A confused mind knows no tranquility. The emotional fallout from uncontrolled desire can not only ruin one's day, but can take down entire families, villages, and nations. Affect leads to effect; emotions have consequences. Yoga advises the restraint of emotion, which can only arise from an honest assessment of one's situation. In the words of Gurāṇi Añjali, understanding leads to acceptance. Acceptance leads to peace, and in peace one finds freedom.

Fourth, Krishna emphatically declares the possibility of freedom through Yoga. If one can reverse the outflows of the senses through managing one's emotions, one can become like a still ocean. One can be wakeful in the midst of ignorance. One can move away from ego fixity and obsession into a state of no ego, no possessions, no lust for the things that bring bondage. The Prajñā Sthiti, the person established in wisdom, becomes godlike, Brahmī Sthiti, and enters the divine abode of Brahma Nirvāṇa (BG II:72), ascending to a heavenly realm characterized as a place where the winds of desire no longer blow.

Religious experience as expressed in this rendering of Yoga does not remove one from the world of the real (*sat*) but from the unreal (*asat*), echoing the Vedic verse quoted at the start of this essay: lead me from the chaos of the unreal (*asat*) to the world of truth and order (*sat*). Arjuna's freedom does not provide an escape from the world but into a place of greater responsibility, with a wisdom that arises from discernment. Arjuna moves away from fear and anger and learns to embrace his action with equanimity.

## 2.1. After the Enlightenment: How to Act with Luminosity, Reason, and Helpfulness

Krishna provides instruction on how to stabilize the body, breath, and mind through concentrated practice in the sixth chapter of the *Gītā*, outlining the practices of Raja Yoga or meditation, leading to a sense of *immediate luminousness*. He teaches devotional practice, Bhakti Yoga, in chapters seven through ten, wherein he instructs Arjuna to view the world as an extension of Krishna's own body, using frequently analogies of light and luminosity. In chapter eleven, where Arjuna witnesses the vast expanse of Krishna's cosmic and eternal form, into which all manifestations eventually are drawn, like moths to a flame, to their death. Here the luminous roars into a state of destructive conflagration, a fire that burns and purifies. In chapter two, Krishna had taught that souls

can never be destroyed. In chapter eleven he shows Arjuna that all bodies can and will be devoured in the jaws of time. This approach to Yoga in many ways elides the distinctions between philosophy, luminosity, and morality, revealing the inescapabilty of darkness and death.

The latter chapters of the Gītā provide a sustained examination of how one can learn to live a life informed and guided by the Yoga of freedom. They make an abiding appeal to adopt an attitude and philosophy of what James calls moral helpfulness. The following passages from chapter 12 (13–19) provide concrete instances of the attitude through which one can manifest moral helpfulness in the name of Yoga:

> 12.13–18 The one beyond hate who shows loving kindness
> and compassion for all beings,
> free from "mine! mine!" and free from ego,
> unruffled in suffering or happiness, patient:
> that Yogi, who is content at all times,
> whose self is controlled, whose resolve is firm,
> … who is of even eye, pure, capable, neutral,
> free from wanting things to be a certain way,
> … who neither elates nor hates,
> neither mourns nor hankers,
> giving up obsession over purity or impurity,
> the same whether with an enemy or a friend,
> the same in honor and disgrace,
> in heat or old, happiness or suffering,
> free from attachment,
> maintaining equipoise when blamed or praised,
> content with whatever happens,
> without fixed abode yet steady minded,
> full of devotion: that one is dear to me. (Translation by the author.)

Krishna encourages Arjuna to adapt a stance of neutrality in the midst of life's vicissitudes. To remain unruffled in the midst of difficulty communicates a stance of ease and peace that can calm the anxious. Similarly, to accept without undue elation life's happy moments can help to prevent an exuberance that can lead to an inevitable let-down.

### 2.2. Philosophy and Moral Assessment through the Three Guṇas:

The three *guṇas* comprise a core teaching of Yoga and Vedānta. They are also at the core of Sāṃkhya philosophy and account for all aspects of potential and kinetic energy (*sukṣma* and *stūla*), subtle and gross, that govern the unmanifest and manifest worlds (*avyakta* and *vyakta prakṛti*). They exist to be witnessed by consciousness (*puruṣa*) and to provide the experience that causes one to seek the understanding that leads to freedom. As introduced in the second chapter of the Gītā (II:45), the *guṇas* describe the changes and fluctuations of states of being, cycling through heaviness and lethargy (*tamas*), action (*rajas*) and buoyant illumination (*sattva*). Krishna advises Arjuna to recognize these qualities and to simply observe that whatever happens, it is "merely the *guṇas* working on the *guṇas*" (III:28). In the fourteenth chapter, Arjuna asks for details, wanting to know the "qualities of the one who has gone beyond the three *guṇas*," asking for a description of the conduct (*ācara*) of the one who goes beyond the three *guṇas* (*trīn guṇān ativartate* XIV:21). Krishna states that such a person not only goes beyond the dualities of the positive and the negative, but transcends the tripartite qualities of "illumination, activity, and delusion (*prakāśaṁ, pravṛttiṁ, moham* XIV:22)" neither hating (*dveṣṭi*) nor desiring (*kāṅkṣati*) their appearance". Knowing that it is "only the *guṇas* working" (*guṇā vartanta ity eva* XIV:23) that person "stands firm, not wavering" (*avatiṣṭhati na iñgate* XIV:23).

Krishna calls for the negation of all dualisms, proclaiming that the Yogi remains the same in the midst of suffering and happiness, love and disdain, blame and praise. However, Krishna also includes an allusion to a threefold distinction that might correlate to the *guṇas* here as well: one is to

have equal regard for a lump of earth, which may refer to *tamas*, a stone, which might refer to *rajas*, and gold, which might correlate to *sattva* (*samaloṣṭāśmakāñcanaḥ*, IV:24). Similarly, Krishna offers one more threefold description of how the person who has transcended the *guṇas* operates: equanimous in honor and dishonor (*tamas*), equanimous whether with friends or enemies (*rajas*), and renouncing all attachment to all undertakings (*sattva*) (XIV:25).

The first three verses of chapter 16 give specific qualities that characterize one with "divine endowment" (*saṁpadam dāivīm*).

> No fear, purity of *sattva*,
> standing persistently in the Yoga of knowledge,
> practicing giving, self-control, and sacrifice,
> study of Self, austerity, appropriate behavior,
> non-violence, truth, no fear,
> giving up attachment, manifesting tranquility, without ill words,
> compassion for beings, without craving,
> kind, modest, and steady,
> vigorous, patient, firm, pure,
> without malice, without excessive pride,
> this, Arjuna, is your birthright,
> this divine endowment. (Translation by the author.)

At the center we find the quality of nonviolence, *ahiṃsā*, the epitome of moral helpfulness. These qualities in the aggregate define *sattva*, the mode of being in the world that brings one closest to the pure witness, the consciousness that gives purpose to all experience. Through careful observance of these behaviors, one moves into the paradigm of the spiritual hero.

### 2.3. Moral Helpfulness and the Sattva Guṇa

Moral helpfulness can be found throughout the seventeenth chapter, which describes many salubrious qualities of the *sattva guṇa*. Krishna praises a reverential attitude (*pūjanam*) toward gods and priests and teachers of wisdom, accompanied with purity, appropriate behavior and comportment, and nonviolence. These are called bodily austerity. Next Krishna describes austerity of speech as calming words that are truthful, lovely, and beneficial, informed by the study and practice of sacred texts. Krishna concludes this triad with a discussion of austerities of the mind, which include cultivation of peace, gentleness, self-restraint, silence, and purity. All these austerities (*tapas*) of body, speech, and mind further emphasize the role of self-development in the practice of moving Arjuna's experience of the world from one of helplessness, despair, and alienation into one of constructive engagement. Chapter seventeen ends by emphasizing the meaning of truth (*sat*) as a state of being (*bhava*) that manifests in laudable actions and words (*karmani* and *śābda*), as well as in sacrifice and austerity (*yajñā* and *tapas*).

These many exhortations urge Arjuna to move toward a place that combines immediate luminousness with moral helpfulness. His descriptions of greater light and lightness are accompanied with various warnings about the results of self-interested action (*rajas*), as well as lethargy and doubt (*tamas*). By the eighteenth and final chapter, three qualities associated with luminosity and morality predominate: sacrifice, giving, and austerity. Arjuna can no longer act from a place of self-interest. Rather than stewing in memories of regret and fear of the future, he is prepared to act and to give freely. Furthermore, he is prepared to give up the fruits of his action, leaving behind all doubt. He has become freed from attachment and ego, steady, resolute, and unconcerned with success or failure (XVIII:26) which allows him to declare "I stand here now with my doubts dispelled, my delusion destroyed. I have regained my memory and am now ready to do what you command" (XVIII:73). Arjuna waged unremitting war on his cousins and suffered in hell as a consequence, having sacrificed even his own well-being for the sake of a higher good. Just as William James talks eloquently and repeatedly about the plight of the sick soul, detailing the sufferings endured by George Fox, Teresa of Avila, and many others, so also Arjuna, as part of his spiritual quest, faced his

own inner fears and doubts in the first chapter of the Gītā, the terrifying face of God in the eleventh chapter, and his own purgation in the depths of hell at the end of the Mahābhārata epic before returning to his divine state.

This single heroic narrative provides a template for religious experience that entails difficulty and suffering, bravery, honesty, and the sustained practice of Yoga in its many forms. In a sense, Arjuna becomes a symbol for every person who seeks solace in the midst of troubles, small or large. The Yogas taught by Krishna, including meditation techniques, discernment, acting without attachment, and devotion, each find usefulness in the story of Arjuna and can be assimilated in their own ways by the modern Yoga practitioner. Before sharing a modern version of how this might take place, attention will now be given to another text that delineates the practice of Yoga, the *Yoga Sūtra* of Patañjali.

## 3. Patañjali's Eightfold Yoga

The Yoga system of Patañjali as given in the 196 statements of the *Yoga Sūtra* (ca. 250 C.E.), defines Yoga as the quelling of thought (*citta vṛtti nirodhaḥ*, YS I:2). Several techniques to attain this state are described, including the eight limbs of Yoga: discipline, observance, ease of bodily movement, control of breath, inwardness, concentration, meditation, and *samādhi*, a state of absorption. The *Yoga Sūtra* of Patañjali along with its accompanying commentary by Vyāsa (ca. 450 C.E.), comprises one of the six core philosophical treatises of Indian thought. It teaches that by gaining inner mastery one can shape one's emplacement in the realm of experience and move towards freedom (*kaivalyam*).

The *Yoga Sūtra* is divided into four chapters, focusing on meditative absorption, the practices required to achieve this state, the powers that consequently arise, and the ascent to freedom. The text begins with a definition of Yoga as quelling the fluctuations of the mind and ends with a description of what Vyāsa calls the liberated soul, freed of afflicted karmas. The Yoga tradition differs from, and remains similar to the five other schools of Indian thought. Unlike the *Brahma Sūtra* of Bādarāyaṇa, a distillation of Vedānta ideas from the Upanishads, Yoga does not claim that the world is in any way illusory and the *Yoga Sūtra* does not use the term Brahman. Like the *Sāṃkhya Kārikā* of Īśvarakṛṣṇa, the *Yoga Sūtra* posits two complementary, eternal principles, consciousness (*puruṣa*) and the events of human experience (*prakṛti*) which are characterized according to three typologies, the pure, active, or lethargic *guṇas*, described in the earlier section on the Bhagavad Gītā. Unlike the *Sāṃkhya Kārikā*, it lists dozens of practices including the efficacy of religious devotion as a possible pathway toward purification. As advocated in the Nyāya system outlined by Gautama, it follows rules of logic. It begins with the premise stated above regarding the quelling of thoughts, proceeds to examine the five categories of thought, and then provides means to purify thought and action. Like the Vaiśeṣika school, it acknowledges the presence of physical realities, and like Mīmāṃsā, Yoga sees benefit in some forms of ritual behavior, particularly in its descriptions of devoting one's attention to a chosen deity or ideal.

The Yoga system also bears traces of influence from Jainism and Buddhism. The first part of Yoga's eightfold path describes the five vows found in the *Ācārāṅga Sūtra*, the earliest extant Jain text (ca. 325 B.C.E.). Like Jainism, it describes karma as multi-colored. It also shares in common with Jainism and Sāṃkhya a concern for the individuality of each particular soul or perspective. Throughout the text it lists terms and practices associated with Theravada and Mahayana Buddhism, including the list of qualities attributed to the liberated Buddhist saint or *arhat* (loving kindness, compassion, sympathetic joy, and equanimity) and markers for spiritual accomplishment including faith, mindfulness, energy, and wisdom as well as stages including the tenth and highest attainment of the Bodhisattva, absorption in the cloud of Dharma. It also seems to engage the Buddhist position on no-self, acknowledging that the ego must be transcended, a key premise of Buddhism, while simultaneously asserting the abiding presence of a witness consciousness, tying Yoga closely to Vedānta and Sāṃkhya.

The second chapter of the *Yoga Sūtra* outlines a threefold method for achieving *samādhi*: rigor (*tapas*), study (*svadhāya*), and dedication to divinity (*īśvara praṇidhāna*). It then describes the five

afflictions that obstruct *samādhi* (*avidyā*, *asmitā*, *rāga*, *dveṣa*, *abhiniveśa*), and describes in detail the first five limbs of Patanjali's eightfold path (*yama*, *niyama*, *āsana*, *prāṇāyāma*, *pratyahāra*). The threefold method, known as Kriyā Yoga, starts with austerity (which in practice often takes the form of regularly fasting and silence) and moves into study of the higher self and dedication to emulating the ideal Yogi. The afflictions to be overcome are ignorance, egotism, attraction to objects of desire, repulsion, and the desire for continuity. Each of these is said to "seed" one's bed of karmas prompting repeated experiences of change and suffering. Patañjali recommends developing discernment to overcome attachment and move into the witness consciousness, deemed to be a state of freedom. By setting aside all the afflictions rooted in ignorance, confusion ceases.

The first two of Patañjali's limbs, the disciplines and observances, require the individual Yogi to abide by a code of ethics and to cultivate positive behaviors, in the style of James's moral helpfulness. As one becomes skilled in nonviolence, enmity ceases in one's presence. By telling the truth, one becomes reliable and one's words hold great sway. By not stealing or even coveting, one finds happiness with what is at hand. By not dissipating one's focus on carnal matters, one gains vigor. By minimizing possessions, one can understand experiences more fully. The positive behaviors to be cultivated include purity, through which one prepares to move into witness consciousness. Through contentment, one becomes abidingly happy. Through austerity one brings the body and senses toward perfection. Through study of the higher self, one begins to emulate the chosen deity. Through dedication to the most accomplished of Yogis, one enters *samādhi.* This process establishes a link between moral helpfulness and immediate luminousness, all in the spirit of philosophical reasonableness.

The remaining passages from the second chapter describe the next three limbs. Yoga postures bring steadiness and ease. Mastery of the inbreath and the outbreath, including the extended hold of each, allows one's innate radiance or immediate luminousness to be revealed. With one's relationship with the world stabilized and purified through the disciplines and observances, and through mastery of body and breath, one then can enter the fifth aspect of Yoga, a place of inward calm.

The third chapter of the *Yoga Sūtra* describes the last three aspects of eightfold Yoga and the powers they generate. Concentration (*dhāraṇā*) leads to meditation (*dhyāna*) and to *samādhi*. From the place of *samādhi*, one re-enters the world with a new skill: the ability to apply focused intention. As one re-engages the world after engaging in times of deep absorption or *samādhi*, the following masteries emerge: knowledge of past and future; ability to understand foreign languages; knowledge of prior births; clairvoyance; ability to remain unseen; knowledge of the time of death; ability to manifest sympathetic joy, and equanimity; physical strength; knowledge of the movements of the sun, moon, and stars; knowledge of the energies of the belly, throat, third eye, head, and heart; ability to experience the bodily feelings of another person; the ability to remain light even in mud or muck; and beauty. This chapter ends with a warning not to become attached to any of these powers, but to always keep the eye on the prize: the state of discernment that releases one from the grip of threefold change of the *guṇas*, summarized above as pure, active, and lethargic.

The fourth and final chapter of the *Yoga Sūtra* elaborates on the operations of karma, reiterates that the state of freedom can never be claimed by the ego, and describes the pinnacle of Yoga as "steadfastness in own form and the power of higher awareness."[8] The key to this state of freedom, the cessation of afflicted action, yields absorption in a cloud of abiding virtue (*dharma megha samādhi*).

Because it includes so many different strands of thought and modes of practice drawn from various Hindu, Jain, and Buddhist traditions, and because it remains open-ended in regard to the choice of deity, or even the necessity to adopt a theological approach to achieve freedom, it became widely read and drew many commentators. It was translated into Arabic in the 10th century by the Muslim philosopher al-Biruni. Since the revitalization of interest in Yoga in the 19th century, it has been translated hundreds of times into many languages, providing a philosophical roadmap for the popular practice of Yoga. Yoga as found in the *Yogavāsiha/Mokopāya* (11th century) emphasizes the centrality of the mind in determining one's place in the world, control of breath, and the elemental meditations. The Jain Yoga of Haribhadra Virahāṅka (6th century) as found in the *Yogabindu* teaches

---

8    (Chapple 2008).

the importance of moving beyond the binding effects of karma, while that of Haribhadra Yākinīputra in the *Yogadṛṣṭisamuccaya* (8th century) emphasizes the many paths of Yoga, correlating Patañjali's eight limbs with the 14 stages of spiritual progress (*guṇasthānas*) delineated in Jainism. The texts of Haṭha Yoga (11th century ff.) provide details on the ascent of energy through the cakras, as well as details on the performance and benefits of *āsanas* and *prāṇāyāma*. The Jain Yoga of the *Jñānārṇava* (11th century) and the *Yogaśāstra* (12th century) includes the Yoga Tantra emphasis on correlations and progressive elemental meditations. In the modern era, the scientific research of Swami Kuvalyananda at Kaivalyadham in Pune informed the Yoga as practiced and taught by Mahatma Gandhi, Swami Sivananda, and Krishnamacharya, who in turn brought the knowledge and practice of Yoga to the masses worldwide, complementing the earlier work of Swami Vivekananda and the philosophical interpretations of Yoga by Sri Aurobindo.

## 4. Yoga in Practice

This special issue of Religions is open to including aspects of religious experience within Hinduism beyond textual studies. Thus far, this article has fallen short of the mark, partly out of a reluctance to "own" my own positionality as a scholar-practitioner and teacher of the Yoga tradition. For five decades, Yoga and meditation have been central to my personal and professional life. For more than a dozen years I studied at Yoga Anand Ashram in Amityville, New York, learning Yoga in theory and practice, simultaneously earning undergraduate and advanced degrees focused on Buddhist and Hindu philosophies and the study of the Sanskrit and Tibetan languages and literatures. Subsequently, as a scholar of religion and a theologian, I have translated and analyzed numerous texts of Yoga, including the *Yoga Sūtra*, the *Bhagavad Gītā*, the *Yogavāsiṣṭha*, as well as Jain Yoga texts including *Yogadṛṣṭisamuccaya*, the *Yogabindu*, and the *Jñānārṇava*. Additionally, I have been indirectly and directly involved in the training of more than a thousand women and men certified by the Yoga Alliance and the International Association of Yoga Therapists, primarily through program development, teaching and supervision at Loyola Marymount University's certificate and degree programs in Yoga Studies, as well at the Hill Street Center in Santa Monica and the YogaGlo online streaming service. Along the way, I have come into the orbit of countless schools of Yoga and meditation practice, including the techniques taught by B.K.S. Iyengar, Pattabhi Jois, Bikram Choudhury, Deshikachar; Swamis Vishnudevananda, Veda Bharati, Chidvilasananda, Adhyatmananda, Bodhananda; the disciples of Swami Lakshmanjoo; Buddhist teachers Philip Kapleau and Trudy Goodman; as well as Jain teachers including Acharyas Tulsi, Mahaprajna, Vidyananda, Siva Kumar Muni, and many others. So, to close this essay, I would like to share a summary of two aspects from a larger Yoga practice that I developed, drawing from these experiences with an eye to how Yoga practice might be grounded philosophically in such a manner conducive to luminosity and moral helpfulness.

### 4.1. A Suggested Daily Practice of Yoga

As we begin the third and final section of this article, the verb mood will move into a form rarely seen in scholarly writing. Academic papers generally employ the indicative mood with an occasional sprinkling of the interrogative, conditional, or subjunctive, generally rendered in the third person to maintain distance and objectivity from the material. However, this next section will switch into a combination of direct command from the perspective of the first person (the author) telling the second person (the reader) how to move the body in a particular sequence of moves that involve breath, verticality, horizontality, motion, and rest. As such, this discourse steps out of a mood and mode of third person remove into a place of direct encounter that holds the possibility of evoking a body-felt experience even in the reading of the material. This next section invites the reader to visualize, to feel, and perhaps to perform.

For anthropologists, this might raise the question of whether the author is taking an emic or an etic approach to the Yoga tradition. Is it possible for a scholar to write about a topic in which one has an investment? Louisa May Alcott's character Jo received sound advice from her professor mentor and her mother in the children's classic *Little Women*: Write what you know. Write from your own

experience.[9] As noted above, my life's work has been as a theologian and philosopher, seeking to develop tools to assist in a search for meaning. This has included the development of curricula for university courses, extension courses, and classes for the general public in the thought and practice of Yoga. Some suggestions are given below for a Yoga class that would be quite different from many of the gym-based exercise versions of Yoga. Two aspects have been identified below from my own learning and teaching of Yoga that could distinguish this form of practice from other popular styles. None of these aspects are "original". They can be found in the Yoga literature, but generally have not been featured in the teaching of mainstream modern postural Yoga. The two are focusing on the five great elements (*pañca-mahābhūta*) and cultivating four positive minds states (*bhāvas*) as delineated in the *Sāṃkhya Kārikā*.

### 4.2. The Five Great Elements: Pañca Mahābhūta

Earth, water, fire, air, and space comprise the basic material and ethereal substances that comprise the human body and the cosmos. Recognition of these five substances while practicing Yoga postures can create a mood of meditative connection. The Bow (Dhanurāsana**)**, followed with the Locust (Śalabhāsana) can serve to call one's attention to the earth and water. While still supine with the stomach and chest upon the floor, one can then rise up into the Full Arm Snake Pose (Nāga) to acknowledge heat and fire, and then into the Bent Elbow Snake pose (Ardha Nāga) to connect with the air. A fifth pose, the Sphinx, wherein one props oneself up on the elbows, can evoke space, completing a fivefold sequence.

Gravity normally pulls the body downward. Every human movement stands in relation to this force. Surrender fully, belly down, to the earth, head turned to the side. In this sequence, to be repeated three or more times, the body rises up away from gravity. Just as the vertical and horizontal movement of the prior sequence inverted and extended the body, the limbs exert an outward and upward movement with similar results.

First, place the chin on the ground. Lift the feet upwards. Reach back and grasp the feet or ankles with the hands and lift the body away from the earth into the Dhānurāsana, the bow pose. With shallow breath, repeat earth, earth earth, *pṛthivī*, *pṛthivī*, *pṛthivī*. Return both arms and legs to the ground and turn the head to one side.

Second, place the chin on the floor and the arms under the thighs, forming a fist with the hands. Lift the legs up into the Śalabhāsana, the Locust pose. Hold for a few seconds and with shallow breath, repeat water, water, water, *jal*, *jal*, *jal*. Bring the legs back to the earth and turn the head to the other side.

Third, place the hands, fingers facing forward, palms down on the floor under the shoulders. Lift up into the Nāgāsana, the cobra pose, with arms extended fully. Hold for a few breaths, repeating fire, fire, fire, *agni*, *agni*, *agni*. Lower the torso to the earth and turn the head to the other side.

Fourth, place the hands once again under the shoulders. Place the toes on the floor, with heels elevated. Lift up into the Ardha Nāgāsana, the half cobra pose, with elbows bent. Visualize the body as if it were a cloud being billowed forward by the wind. Hold this posture for a few seconds, repeating air, air, air, *vāyu*, *vāyu*, *vāyu*. Lower to the earth and turn the head to the other side.

Fifth, elevate the front of the body, with elbows on the floor, entering the Sphinx Pose. Gaze forward as if looking into the vast sands of the Sahara. Repeat space, space, space, *ākāśa*, *ākāśa*, *ākāśa*. Lower to the earth and turn the head to the other side.

Repeat the sequence as above, moving backward from the elements in a movement known as *pratiprasava*, this time evoking the subtle elements or *tanmātras* and their connection with the sense organs, the *buddhīndriyas*. While in Dhanurāsana, reflect on the process of smelling with the nose, *gandha* (fragrance) known through *nasa* (the nose). While in Śalabhāsana, reflect on the process of tasting with the mouth, *rasa* (flavor) known through the tongue, lips, and palate (*mukha*). While in Nāgāsana, reflect on the process of seeing with the eyes, apprehending *rūpa* or form with the eyes (*akṣa*), rotating the eyes first in one direction and then the other. While in Ardha Nāgāsana, feet

---

9   (Alcott [1869] 1987).

perpendicular to the ground, reflect on feeling or *sparśa* through the largest organ, the skin or *tvak*. While in the Sphinx Pose, bring attention to the ears or *karṇa*, the gateway to sound or *śabda*.

In the third repetition, focus in turn on the correlations between the Dhanur Pose and the lifting of the anus away from the force of gravity; in Śalabhāsana, the lifting of the genitals away from the earth; in Nāga, the power of the hands as they push against the earth; in Ardha Nāga, the legs as they push into the earth; in Sphinx, bring attention to the voice, the throat, the larynx. These motor functions allow full engagement with all aspects of the manifest world.

This sequence completes mindfulness of the twenty *tattvas* that connect the body and the world: the five gross elements of earth, water, fire, air and space; the five subtle elements that allow smelling, tasting, seeing, touch, and hearing; the five sense organs of nose, mouth, eyes, skin, and ears; and the five motor capacities of evacuating, allowing the passage of water, grasping with the hands, walking with the arms and feet, and speaking with the voice.

### 4.3. The Four Bhāvas of Positivity

In the *Sāṃkhya Kārikā*, Īśvarakṛṣṇa emphasizes the disposition of one's emotional outlook in the determination of experience within the world. The intellect, according to this philosopher, finds itself constituted internally, awaking, as it were, to a world inseparable from one's emotional landscape. The term for intellect, Buddhi, derives from the verb root Budh, which means "awaken". If one awakens into weakness, attachment, ignorance, and viciousness, trouble will result. In the sequence that follows, one trains to engage the reverse.

Sit up straight extending both legs to the front. Bring the right foot inside the left thigh. Reach up toward the sky and extend outward, bringing the head toward the knee and grasping the big toe or foot if possible. Move into Paścimatānāsana**.** Speak the positive quality (*bhāva*) that indicates empowerment, *aiśvarya*. Release both feet forward. Bring the left foot inside the right thigh. Reach upward and then bring the head toward the right knee, grasping the big toe or foot. Speak the positive quality for non-attachment, *virāga*. Bring both feet forward. Stretch upward and outward, bringing the head toward the knees, grasping the big toes or feet if possible. Speak the word for liberative knowledge, *jñāna*. Release and bring the feet out in front once more. Bring sole to sole, moving the heels toward the perineum, moving into butterfly pose, Baddha Koṇāsana. Speak the word *dharma*. These four terms indicate the positive attitudes and states of being (*bhāvas*) that can be cultivated through yogic intention: *aiśvarya*, *virāga*, *jñāna*, and *dharma.* Repeat twice more, utilizing your own phrases for each positivity. The *bhāvas* determine one's outlook and attitude, allowing ascent into higher states of awareness and, in the words of William James, moral helpfulness. Their description can be found in the *Sāṃkhya Kārikā*, 23 and 44–46.

These two practices each serve to reposition one's sense of self away from mindless repetition of past actions into their epoche or suspension and entry into a time and space of purposeful intent. In a sense, these moments and movements bring forth the sort of reversal described in the *Bhagavad Gītā*:

> The Blessed One said:
> They speak of the changeless aśvattha tree,
> its roots above, its branches below.
> Its leaves are the Vedic hymns.
> The one who knows it knows the Vedas.
> Its branches stretch below and above, nourished by the *guṇas*.
> Its sprouts are the sense objects.
> In the world of people,
> it spreads out the roots that result in action (BG 15:1–2, translated by the author).

This metaphor suggests that actions in the world can be called back into an unmanifest space, a place of silence, not unlike the process described in the second chapter of the Gītā wherein the yogi remains unruffled in the midst of change. By focusing on the elements and the interlinkage between the senses and the objects and actions of the external world, one can develop mindfulness that allows appreciation and, when needed, a skillful remove. By cultivating emotions and attitudes of positivity

as recommended in the *Sāṃkhya Kārikā*, one can create predispositions that will help overcome the inevitable difficulties that arise in the course of daily life.

These two examples of an integrated, thoughtful Yoga practice, seek to link movement with higher intent. While perhaps not hard-wired into most Yoga experiences in this exact shape and form, awareness of the five elements and the concept of improving one's disposition were undoubtedly well known to the originators of modern traditions of Yoga, many of whom were mentioned earlier. A bit like the children's game of telephone, where a phrase whispered from one to another will be altered by the time it reaches the other side of the room, yes, Yoga has undergone many changes in the processes of translation and reception. However, as Andrea Jain has noted, this adaptability has been a hallmark of the Yoga tradition over the course of several centuries.[10]

Many people associate Yoga with physical flexibility and with agnosticism when it comes to things religious or philosophical. The practices of moving toward greater lightness and increased virtue emphasized in this article serve to complement the enhanced physicality and philosophical openness of Yoga practice. Not only can it make one's body limber and strong, Yoga can effect positive change in personality.

Does the fetishization of the physique benefit a rigorous Yoga practice or detract from what some might perceive to be its "pure" message and intent? Perhaps. Can obsession with form cause a destabilization of the body and emotion? Certainly. Additionally, the potential shadow side of Yoga teacher power dynamics must be acknowledged and critiqued. Amanda Lucia insightfully analyzes the ways in which a Yoga teacher can be deified, sometimes setting the stage for scandal and abuse.[11] These all too common and unfortunate occurrences violate the precepts of Yoga which are grounded in non-violence (*ahiṃsā*) and truth (*satya*). For those who adhere to the vision of Yoga that enhances self-worth and self-respect leading toward "the light," Yoga can be an important gateway to places of luminous encounter, philosophical insight, as well as kind and helpful actions.

## 5. Conclusions

This article opens and closes with the quotation of mantras, words from the Sanskrit language that establish a mood of connection. The opening verses beckon to the sun and the inner light. The paradigm of the Yogi as described in the Gītā describes a process of inner stabilization as a ground for the Yoga experience. The *Yoga Sūtra* outlines a reciprocity between the cultivation of self and one's relationship with the world. Thoughtful daily practice of Haṭha Yoga has been described here as well, through movements and intentions that connect with the elements and uplift one's attitude and mood. The combination of all these aspects of Yoga enhance the possibility of positive transformation. Yoga makes a call, a suggestion that purpose and meaning in life can be found within and without. Through stabilizing one's body, emotions, and thoughts, one can cultivate states of luminosity, insight, and helpfulness, embracing an integrated sense of religious experience.

**Funding:** This research received no external funding.

**Acknowledgments:** 

**Conflicts of Interest:** The authors declare no conflict of interest.

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
