# Peer review of "Religious Experience and Yoga"

_religions, doi:10.3390/rel10040237_

Round 1
Reviewer 1 Report
This article is excellent. It is clearly and engagingly written and the topic is absolutely central to the theme of the special issue of which it will be part. I recommend no changes and that it be accepted in its current form.
Author Response
Thank you!
Reviewer 2 Report
Religious Experience and Yoga
13. There may be room to differentiate the Yoga school from the Vedic one as their underlying assumptions are different. At least indicate that you are aware of that or say something about the link between yoga and Vedic Brahminism.
45. "In Hindu Yoga traditions"; as opposed to which non Hindu yoga traditions? Buddhist? Jain? Please insert a note indicating what you mean.
59. "This seminal work in many ways places Yoga at the nexis of ways to conceptualize the entire conceptual realm of having a religious experience, in terms of both process and actualization." The sentence is cumbersome.
74. "four forms of Yoga articulated in the Bhagavad Gῑtā: discernment or Jñāna Yoga, action or Karma Yoga, devotion or Bhakti Yoga, and meditation or Raja Yoga." What about Hatha Yoga?
80. "In various ways, Krishna instructs Arjuna on the complexities of moral helpfulness, specifying that Karma Yoga, with its sense of aplomb, will see one through even the most difficult of tasks, and that it is possible to hold to one’s dignity whether faced with humiliation or glory. " not quite sure what you are talking about; it least provide the relevant gita verse of verses.
88. "Gandhi landed upon the Bhagavad Gītā". Landed? A rather strange description.
90." He sought solace particularly in the last eighteen verses of the second chapter (59-72)". 72-59 is not 18. It should be 19 and it is 54-72.
116. May we ask which verses are you quoting? An endnote is warranted here. It seems more like an undergraduate paper than a well edited one.
130. Same; quote is missing.
148. "Krishna urges one to “dial it back,” to recognize that a sense object does not exist before the sensory organ (indriya) “lands” upon it, seizes it, and makes it real." An obscure sentence.
170. echoing the Vedic verse quoted at the start of this essay
180 "This approach to Yoga in many ways elides the distinctions between philosophy, luminosity, and morality, revealing the inescapabilty of darkness and death." Rather obscure and an explanation is required.
189 "what James calls moral helpfulness". A least a footnote is required including a proper citation and some explanation.
Author Response
A corrected version has been prepared and submitted. Thank you for your careful read. On a couple of points, there was no change.... Hatha Yoga as a textual tradition emerged many centuries following the Bhagavad Gita. Moral helpfulness is defined by William James right at the start of the article as an after-effect of religious experience.